# Experimental Investigation of Low-Frequency Distributed Acoustic Sensor Responses to Two Parallel Propagating Fractures [note 1]

**DOI:** 10.3390/s24123880

**Published:** 2024-06-15

**Authors:** Teresa Reid, Gongsheng Li, Ding Zhu, A. Daniel Hill

**Affiliations:** 1Harold Vance Department of Petroleum Engineering, Texas A&M University, College Station, TX 77843, USA; 2Chevron, 1600 Smith Street, Houston, TX 77002, USA; 3Pioneer Natural Resources, 777 Hidden Rdg, Irving, TX 75038, USA

**Keywords:** fiber optic sensing, distributed acoustic sensor, multistage hydraulic fracture stimulation, fracture diagnosis

## Abstract

Low-frequency distributed acoustic sensing (LF-DAS) is a diagnostic tool for hydraulic fracture propagation with far-field monitoring using fiber optic sensors. LF-DAS senses strain rate variation caused by stress field change due to fracture propagation. Fiber optic sensors are installed in the monitoring wells in the vicinity of a fractured well. From the strain responses, fracture propagation can be evaluated. To understand subsurface conditions with multiple propagating fractures, a laboratory-scale hydraulic fracture experiment was performed simulating the LF-DAS response to fracture propagation with embedded distributed optical fiber strain sensors under these conditions. The experiment was performed using a transparent cube of epoxy with two parallel radial initial flaws centered in the cube. Fluid was injected into the sample to generate fractures along the initial flaws. The experiment used distributed high-definition fiber optic strain sensors with tight spatial resolutions. The sensors were embedded at two different locations on opposite sides of the initial flaws, serving as observation/monitoring locations. We also employed finite element modeling to numerically solve the linear elastic equations of equilibrium continuity and stress–strain relationships. The measured strains from the experiment were compared to simulation results from the finite element model. The experimentally derived strain and strain-rate waterfall plots from this study show the responses to both fractures propagating, while the fracture at the lower position took most of the fluid during the experiment. Interestingly, a fracture first began propagating from the upper flaw of the two flaws, but once the lower fracture was initiated, it grew much faster than the upper fracture. Both fibers were intercepted by the lower fracture, further verifying the strain signature as a fracture is approaching and intersecting an offset fiber.

## 1. Introduction

Low-frequency distributed acoustic sensing (LF-DAS) is a diagnostic tool for hydraulic fracture propagation in multistage fracture treatments. To monitor fracture development from a treatment well where high-pressure injection induces hydraulic fractures along the well, a far-field monitoring well is required. A fiber optic sensor is installed on the monitoring well. When fractures approach the monitoring well, the stress/strain field surrounding the fractures varies [1,2,3,4]. This change can be registered by the fiber optic sensor through low-frequency distributed acoustic sensing (LF-DAS). In the field, the frequency band used for this diagnosis is usually <1 Hz. The application of fiber optic LF-DAS for fracture diagnosis is shown in Figure 1 [5].

The study of LF-DAS in field cases has revealed characteristics of fractures and their geometry. Published works have related LF-DAS signatures to fracture driven interactions at wellbores offset to treatment wells [6,7,8,9,10]. Offset well LF-DAS data display narrowing regions of extension surrounded by areas of compression, indicating a fracture is approaching the offset well. During fracture opening, there is a consistent area of extension, and upon fracture closure, the polarity of the strain response at the fracture plane flips from extension (red) to compression (blue) as shown in Figure 2.

To understand subsurface conditions with multiple propagating fractures, a laboratory-scale hydraulic fracture experiment was conducted using an established workflow [1]. The previous work consisted of laboratory experiments simulating single orthogonal fracture propagation in a linear elastic medium (epoxy) with embedded fiber optic cables measuring strain. The initial work developed a method to estimate the fracture radius and distance of a single fracture’s front to the nearest fiber [5]. The method was validated with laboratory experiments and applied to field cases to evaluate accuracy of the method. Laboratory-scale experiments validated the converging extension pattern surrounded by compression to indicate a fracture intersection with a fiber. A zero-strain model was developed to estimate fracture geometry from the location of zero-strain along a fiber optic sensor [5]. A follow-up study investigated the shear effect on DAS measurements in the same laboratory setup with an angled fracture under uniaxial compressional load. The zero-strain location model tracking fracture radius was proved to be able to accurately predict the fracture radius and distance to the fracture front when given the strain data. The strain response from the fiber that was intercepted by the fracture suggested that a fracture under shear and normal force has an asymmetrical LF-DAS response over the location of the fracture [11].

In this paper, a similar workflow was applied to understand the stress impact on two parallel fractures being initiated at the same time while fiber optic strain responses were recorded. The ability to study LF-DAS with multiple propagating fractures can reveal insights into how fiber optic data can be utilized to diagnose multiple fracture propagation and interference between fractures.

## 2. Theoretical Basis of the Research

The linear elastic theory consisting of equilibrium, continuity, and linear stress–strain relationships was applied to model the experimental domain. The domain was a cube with a dimension on each side of 8 inches. The basic equations are provided in Equation (1) for cylindrical coordinates [12]:(1)$$\frac{\partial \sigma_{r}}{\partial r}+\frac{1}{r}\frac{\partial \tau_{r\theta }}{\partial \theta }+\frac{\partial \tau_{rz}}{\partial z}+\frac{\sigma_{r}-\sigma_{\theta }}{r}=0\;\frac{\partial \tau_{rz}}{\partial r}+\frac{1}{r}\frac{\partial \tau_{\theta z}}{\partial \theta }+\frac{\partial \sigma_{z}}{\partial z}+\frac{\tau_{rz}}{r}=0\;\;\frac{\partial \tau_{r\theta }}{\partial r}+\frac{1}{r}\frac{\partial \sigma_{\theta }}{\partial \theta }+\frac{\partial \tau_{\theta z}}{\partial z}+\frac{2\tau_{r\theta }}{r}=0$$
where σ and τ are normal and shear stresses in a cylindrical coordinate system defined by r, θ, and z. Sneddon solved this system of equations for the displacements and stresses due to a penny-shaped crack in an infinite linear elastic medium [13]. These equations can also be solved numerically using the finite element method. To compare with Sneddon’s theory, we utilized a commercial software to model the stresses and strains in an 8-inch epoxy fracture specimen (Dassault Systems, Vélizy-Villacoublay, France, 2018). We refer to this approach as the finite element solution in Section 5.

For a penny-shaped crack propagating parallel to the sides of the cube, the relationship between fracture volume, *V_f_*, net pressure, *P_net_*, and the crack radius, *R*, to the rock mechanical properties can be expressed as follows [14]:(2)$$V_{f}=\frac{16P_{net}R^{3}}{3E^{\prime }}$$
(3)$$P_{net}={\left(\frac{\pi^{3}K_{IC}^{6}}{12E^{\prime }V_{f}}\right)}^{\frac{1}{5}}$$
(4)$$R={\left(\frac{3V_{f}E^{\prime }}{8\sqrt{\pi }K_{IC}}\right)}^{\frac{2}{5}}$$
where *E*′ is the plane strain modulus and *K_IC_* is the critical stress intensity factor. Sneddon provided an expression for the displacement, *u_z_*, at any location in an infinite medium due to a radial crack in the direction normal to the plane of the crack, *z* [14]. By numerical differentiation of the displacement, the axial strain, $\varepsilon_{z}$, normal to a radial fracture can be computed.
(5)$$\;u_{z}=\frac{\left.-4P_{net}R(1-\nu^{2}\right)}{\pi E}\int_{0}^{\infty }\left(1+\frac{\zeta \eta }{2\left(1-\nu \right)}\right)\left(\frac{cos\eta }{\eta }-\frac{sin\eta }{\eta^{2}}\right)e^{-\zeta \eta }J_{0}\left(\rho \eta \right)d\eta$$
(6)$$\varepsilon_{z}=\frac{\left.-4P_{net}R(1-\nu^{2}\right)}{\pi E}\frac{d}{dz}\int_{0}^{\infty }\left(1+\frac{\zeta \eta }{2\left(1-\nu \right)}\right)\left(\frac{cos\eta }{\eta }-\frac{sin\eta }{\eta^{2}}\right)e^{-\zeta \eta }J_{0}\left(\rho \eta \right)d\eta$$

Neglecting strain transfer effects from the fracture medium to the fiber, Equation (6) can be used to estimate the measured strain on a fiber optic cable offset normal to the fracture plane. The strain depends on the fracture radius, net pressure, the Young’s modulus, E, and Poisson’s ratio, ν. The variable *η* is a parameter from the Henkel transform used in the solution of the differential equations. The dimensionless depth and radial coordinates, ζ and ρ, respectively, are defined as follows:(7)$$\zeta =\frac{z}{R}$$
(8)$$\rho =\frac{r}{R}$$

Sneddon’s semi-analytical equations were solved using numerical integration and differentiation. The results from this approach are referred to as Sneddon’s solution in Section 4. The solutions from Sneddon and the finite element approach were calibrated against each other when analyzing the results. Sneddon’s solution provided a more efficient means for modeling the field response compared to finite element modeling. On the other hand, the finite element model handled the boundary conditions of the 8-inch fracture specimens more appropriately. Sneddon’s solution was adjusted to more effectively model the strains on the fiber in the experiments.

The experiments were designed to help understand LF-DAS measurements at monitoring wells while the treatment well is fractured. For convenience in applying models to field applications, we introduce dimensionless parameters in terms of well spacing:(9)$$z_{D}=\frac{z}{\sqrt{∆l^{2}+∆h^{2}}}$$
(10)$$R_{D}=\frac{R}{\sqrt{∆l^{2}+∆h^{2}}}$$
where ∆l is the horizontal well spacing between the treatment and monitor wells, and ∆h is the vertical offset. Equations (9) and (10) are useful for upscaling laboratory results to field scale. In the lab experiments, ∆h is zero, and $z_{D}=z/∆l$.

For a radial fracture, there is a one-to-one correspondence of zero-strain location to fracture radius where the radius can be estimated with a simplified version of Sneddon’s solution [13], and the zero-strain location is dependent only on the fracture radius, the monitor well vertical and horizontal are offset from the treatment well (referring Figure 1), and a representative Poisson’s ratio of the formation is taken.
(11)$$\varepsilon_{z}=\frac{-4P_{net}R}{\pi E^{\prime }}f\left(\nu ,z_{0D},R_{D}\right)=0$$
where $\varepsilon_{z}$ is the axial strain at any location in an infinite medium for a radial crack in the direction normal to the plane of the crack as a function of fracture net pressure, $P_{net}$; and with fracture radius, R; Poisson’s ratio, v; Young’s Modulus, E; dimensionless zero-strain location, $z_{0D}$; and dimensionless crack radius, $R_{D}$.

The dimensionless crack radius, *R_D_*, is a function of Poisson’s ratio,
(12)$$R_{D}=\frac{-b-\sqrt{\left.b^{2}-4a(c-z_{0D}\right)}}{2a}$$
where the constants, a, b, and c are approximated in Table 1.

The model result estimates the distance from the monitoring well to the fracture front, $d_{f}$, which is dependent on the lateral and vertical offset between the fiber and monitor well, ∆l and ∆h, respectively, and $R_{D}$.
(13)$$d_{f}=\sqrt{{∆l}^{2}-{∆h}^{2}}\left(1-R_{D}\right)$$

The experiments led to the development of the zero-strain location method which can dynamically estimate fracture propagation. The experiments determined that the location of zero-strain has a one-to-one correspondence with fracture radius [5]. The work was validated with experimental data and field cases. These experiments laid the groundwork for how LF-DAS can be studied in the laboratory and scaled to field applications to detect approaching fractures in a far-field well.

## 3. Experimental Approach

In this study, to continue from the single orthogonal fracture experiments, we introduced a two-fracture system. A schematic representation of the lab-scale hydraulic fracture experiment is shown in Figure 3. An 8-inch cube of epoxy with known mechanical properties was used in the experiment to represent formation rock. Two flaws were embedded at the center of the block, each offset 1.3 inches from the middle of the cube, resulting in a 2.6-inch fracture spacing. The flaw assisted the initiation of the two fractures. A syringe pump injected dyed water into the center of the epoxy fracture specimen to propagate the fractures. The flow rate was controlled through the pump controller on the syringe pump. No external load was applied during this experiment. The pressure during injection was recorded from a pressure transmitter and the fracture geometry was recorded with a video camera at a speed of 30 frames per second. Multiple camera angles were utilized to determine which fracture was propagating. Two high-definition strain sensors (at the monitoring well) were used to measure the strain along offset locations, 2 inches from the center of the injection tubing (as the treatment well) and 1 inch from the radial fracture initial flaw.

### 3.1. Fracture Specimen

The fracture specimen was created within a plywood mold lined with polypropylene sheathing tape. The EcoPoxy Flowcast resin and hardener system was used to create the epoxy cubes representing the reservoir in a LF-DAS application. The temperature of the laboratory during curing and the experiment was maintained at 72 °F to keep mechanical properties consistent with the previous experiments. The Young’s Modulus for the epoxy sample was 354,000 psi, and Poison’s Ratio of 0.35 was used for the study [5]. As per the specifications of the epoxy used, to ensure the consistency of the sample properties, the epoxy sample was constructed as a six-layer epoxy cube with each layer designed for a cured thickness of 1.33 inches. Each layer cured for 72 h before another layer was poured. The injection test was performed 7 days after the final layer was poured.

After the second and fourth layers were cured, the 1-inch diameter tape was centered in the middle of the cube on top of those surfaces. After layer five was poured, a ½-inch diameter hole was drilled 1.5 inches into the epoxy through the middle of the fracture. A ¼-inch stainless steel tubing was set in the hole with quick-setting epoxy. An O-ring was set on the injection tubing 1 inch above the upper initial flaw to preserve connectivity between the end of the tubing and the initial flaw. Figure 4 covers a schematic representation of the sample in addition to two photographs of the resulting specimen prior to the injection test.

### 3.2. Strain Measurements

Strain measurements were recorded from the two embedded fiber optic cables offset 2 inches from the center of the two fractures in opposite directions. The measurements were recorded by an ODiSI 6100 interrogation system [15]. High-definition strain sensors with a spatial resolution of 0.65 mm were used at a sampling frequency of 6.25 Hz. Table 2 highlights the operational parameters of the interrogator unit utilized.

Two video cameras were used to take images as the fractures were propagating. These images were used to estimate fracture geometry as a function of injection time. They also illustrated the growth and competition between the two fractures during the experiment.

### 3.3. Experimental Procedure

The experiment was conducted with an injection rate of 0.25 mL/min. There was a pressure transducer at the injection pump that also recorded pressure data during the experiments. Once the DAS and pressure measurements were stable (at 0 second in the results reported), fluid injection began, with a rate of 0.25 mL/min. The injection lasted for 600 s. When one of the fractures broke through the specimen, the injection was completed. The injection was paused for 21 s before reversing the flow at 0.1 mL/min suction rate for 1080 s to release the pressure in the system.

## 4. Results and Discussion

The experimental results consist of fracture geometry images, strain and strain rate waterfall plots, and zero-strain rate method analysis. To confirm and better understand the experimental results, the results of a numerical model solving the linear elastic equations of equilibrium continuity and stress–strain relationships are also presented. The observations and findings will be discussed in Section 5.

### 4.1. Fracture Geometry

The fracture geometry captured during the experiment by the cameras is depicted in Figure 5. There are six sets of images in Figure 5. In each set, the top image is a top-view of the fractures, and the bottom image is the angled side-view of the fractures. It is easier to mark the fiber location in the top views, but the fracture growth is more obvious in the angled side-views.

Figure 5 shows the sequence of the fracture propagation. At the initial condition, the locations for both fractures were just the artificial flaws (red-color, Figure 5a). As injection started, the top fracture was filled first with the injection fluid which was blue-colored (Figure 5b). As the top fracture propagated slightly, the bottom fracture started taking the fluid (Figure 5c). Once the bottom fracture started propagating, it grew faster than the top fracture (Figure 5d–f) and dominated the fracture propagation.

Because the injection was from the top, it gave an advantage to the top flaw to initiate the fracture first. But it seems that it was easier to grow the bottom fracture once the system was established. The fractures were both radial shaped in the homogeneous medium. The camera angle in the images above did not change; however, the cube started to rotate in a clockwise direction as the injection continued.

Figure 6 displays the top-view pictures for the fiber interception during the experiment. The bottom fracture hit the north fiber slightly earlier than it intercepted the south fiber at 315 s of the experiment (Figure 6a). At 393 s of the experiment, the bottom fracture intercepted with the south fiber (Figure 6b). As the fracture propagated, the fracture grew, passing both fibers (Figure 6c). At 600 s, the bottom fracture broke through the specimen, and the injection was terminated (Figure 6e).

The controlling factor of the competition between the two fractures during the injection is believed mainly coming from the capped end of the injection tubing. In theory, the injection fluid hits the closed end and generates additional momentum to fracture propagation. The bottom fracture is closer to the end of the injection tube, taking advantage of fracture propagation. This only happens in ideal experimental conditions, however. In reality, fracture propagation can be affected by many other factors, including well completion structures and formation properties. Heterogeneity of rock mechanical properties is one of the most significant controlling factors for multiple fracture competition and propagation. Observations from the experiments indicated that fracture propagation is extremely sensitive to the surrounding conditions.

### 4.2. Strain and Strain Rate Behavior

Figure 7 presents the strain results from the north fiber, and Figure 8 is for the south fiber. These figures include (a) experimentally collected strain, (b) derived strain rate, and (c) pressure (left) and rate (right) data during the experiment.

The waterfall plots (color-contour plots) of strain and strain rate for the experiment that are comparable to the field records were generated as functions of time (along *x*-axis) and distance from the fracture plane (along *y*-axis) for each fiber embedded in the cube. The experimental results were plotted in strain waterfall plots where positive strain (or extension) was represented in warm-color (red), and the cold-color (blue) represented negative strain (or compression). The strain rate was also plotted. Due to the 6.25 Hz sampling rate of the equipment, the strain rate measurements were very noisy after the data processing. To overcome this, the strain-rate was down-sampled by averaging the strain change over 1-s intervals to reduce the noise in the data. The black lines represent times of fracture growth.

The strain waterfall plots reflect propagation of both fractures starting around 200 s, where the strain increases in magnitude at both fractures’ plane locations (at 2.6 inches and at 5.3 inches from the top of cube for the top and bottom fracture, respectively). As the bottom fracture propagates, the classical characteristics of fracture approaching and intercepting are recorded by the fiber. At 454 s, there is a cone-shaped strain pattern that concaves to the time that the fracture intercepts the fiber. After the fracture intersection, there are narrow zones of compression. The strain responses between 3 and 5 inches from the top show the strongest magnitude of extension. The red dashed line on the strain plot (Figure 7a) indicates the moment that the fracture hits the fiber. After the fracture intersection at the depth of the bottom fiber near 5 inches, the extension response exceeds the system measurement capability of the optical interrogator unit and therefore no data is available. Figure 7b highlights a similar cone-shaped pattern of extension leading up to the intersection of the fracture with the north fiber; a compressing signature is also observed post fracture intersection on the strain rate plot.

Another interesting observation from the strain plot is the less significant strain response above the main feature (at about the 2 inch location on the vertical axis). It is believed that the minor response is from the top fracture, which was never developed enough during the injection.

A previous study showed that fracture propagation is in a step-change manner. Every time the fluid breaks the medium and the fracture grows a little, there is a sharp decrease in the pressure curve [5]. This was observed also in this experiment. Figure 7c shows the pressure record for the experiment. The early small pressure drops were most likely due to the top fracture growth. Once the bottom fracture took over the propagation, clear pressure decreases were observed. The black lines in Figure 7 indicate the corresponding times when the fracture propagated further.

Given the radial geometry of the fractures observed during the experiment, the north and south fiber responses are comparable. There is a strong extension response observed in-between the two fractures, and the bottom fracture experiences more strain than the upper fracture given the bottom fracture intercepted the south fiber. Similarly, there is a cone-shaped pattern in both the strain and strain rate waterfall plots leading to the time of bottom fracture intersection of the south fiber at 501 s. Additionally, compressional lobes around the depth of fracture intersection are present following the inception of the fracture with the fiber.

Comparing with the previous experiment, the strain rate waterfall plots for a one-fracture system and two-fracture system are shown in Figure 9. The top picture is for the one-fracture system, and the bottom one is for the two-fracture system. For the single fracture, the strain rate waterfall plot is more symmetrical than the two-fracture system. In the ideal homogeneous medium, this difference is caused by the stress interference while both fractures begin to propagate.

### 4.3. Application of Zero-Strain Location Method to Experimental Results

The zero-strain location method to detect the fracture front [5] was applied to the two-parallel-fracture experimental results. Figure 10 is the strain waterfall plot for the north fiber measurement during the experimental waterfall plot (Figure 7 top picture), in which the black crosses represent the locations where the strain value was nearest to zero. As indicated in the color scale legend to the right, these zero-strain locations are represented by the green color in the waterfall plot. The asymmetric shape of the zero-strain locations from above and below the bottom fracture plane shows the influence of increased strains from the top fracture propagation. Therefore, the depths at the zero-strain locations from below the fracture plane are used in the zero-strain location method as inputs to the model for predicting the effective fracture radius and distance to the fracture front.

The zero-strain location method was applied for the zero strains below the bottom fracture plane for the two-fracture experiment. Figure 11 and Figure 12 show the results of the zero-strain location method. The blue crosses represent the modeled effective radii (Figure 11) and the distance from the fiber location to the fracture front (Figure 12) results against the experimentally measured effective radii and distance to the fracture front, respectively. The blue crosses are the model results from using the zero-strain locations below the bottom fracture plane to avoid the impact from the top fracture propagation. The application of the zero-strain location method shows agreement with the measured results (in black line) for the zero-strain locations available before the fracture interception. The results between 250 s and 283 s suggest that the zero-strains method from below the fracture plane overestimates the effective fracture radius while underestimating the distance to the fracture front. The accuracy of the zero-strain location method increases as the fracture propagates closer to the north fiber between 283 s and 450 s. In this case, about 80% of the data can be used to detect the fracture front. This methos provides an early detection of fracture interception to nearby wells in the field.

## 5. Finite Element Modeling

The experimental conditions were also numerically simulated by solving the linear elastic equations of equilibrium continuity and stress–strain relationships. A finite element model GEO3D [8] was used to solve the total stress, strain, and displacements within the simulation domain. This model was validated against the Sneddon’s solution for calculating strains from a radial fracture [16]. Due to the symmetrical configuration of the experiment setup, the finite element model was applied with a domain of one-half of the 8-inch epoxy cube. This enabled the prediction of strain values at the location of the fiber. Figure 13 displays the simulation system. The 8-inch cube of the experimental specimen with the two flaws to initiate fractures is shown in Figure 13a. In Figure 13b, the flaw location and the distance between the flaws are marked. Recall that each dashed line on Figure 13b marks a 1.33-inch increment in distance from the top. The simulation domain with two parallel fractures and the north fiber is shown in Figure 13c. A mesh generator customized the mesh design based on the fiber location, wellbore configuration, and fracture geometry. To enhance the accuracy of results from the finite element model, we implemented a highly refined mesh around the center of the wellbore and the fiber. For this study, we employed 20-node hexahedral elements to mesh half of the 8-inch epoxy cube (Figure 13). The size of the simulation domain was 8 × 4 × 8 inches. The Poisson’s ratio of 0.35 and Young’s modulus of 355,342 psi were used for the numerical simulation based on the epoxy property specifications.

Figure 14 shows the simulation result at 300 s of injection. Figure 14a presents the relative positions of the two parallel fractures and north fiber from the experiment. At 300 s into the experiment, the top fracture radius was 1 inch and the bottom fracture radius was 1.8 inch. The measured pressure of the fracture was 530 psi. Figure 14b provides the finite element modeled strain response (in dashed-blue line) compared to the measured strains from the experiment result (in black line). The simulation results are consistent with the experimental measurements.

Between the two fracture planes, there is a continuous extensional strain response observed in both measured and simulated results (Figure 14b). This leads to a broader region of extensional strain than in the single-fracture case. Notably, the bottom fracture experiences a greater degree of extensional strain compared to the top fracture, given that the bottom fracture propagates closer to the fiber. In the experiment, variations in epoxy properties along the fracture plane, such as stress and mechanical properties, might be slightly non-uniform. However, the model assumes isotropic and homogeneous conditions for the entire domain. The primarily different input is in the fracture net pressure. Based on the experiment, the height of the fluid column above the bottom fracture is greater than the height above the top fracture, leading to a higher hydrostatic pressure at the bottom fracture location. Consequently, the bottom fracture propagates more easily than the top fracture in the experiment, as it experiences a higher net pressure. This observation indicates that fracture propagation is sensitive to the net pressure condition. In real life, combined with the heterogeneity of formation rock properties, interpretation of strain measurements can be extremely challenging.

The finite element model with half of the 8 inch cube domain effectively simulates the measured strains before the fracture intersection. Although the absolute magnitude of the measured strains differs from the simulated strains at some locations, the overall strain response aligns accurately along both fracture planes. In the context of this study, when a material behaves in a linear elastic manner, the experimentally measured strains should precisely match the strains predicted by the linear elastic model. However, according to Leggett et al. [5], the same epoxy material used in this experiment demonstrates viscoelastic properties, which introduce nonlinearity into the stress–strain relationship. Therefore, the strains measured from the epoxy diverging from the theoretical strains is not unexpected based on the linear elastic assumption. Despite the effect of nonlinearity, the finite element model remains a valuable tool for predicting the strain response for the two-parallel-fractures case, and any disparities between the model’s predictions and the actual measurements are within an acceptable margin of error.

The study presented in this paper illustrates that when more than one fractures propagate and approach the monitoring well, the pattern of strain waterfall plot becomes asymmetrical. This observation can be used in field analysis to identify multiple fracture systems. Keep in mind that asymmetrical characteristics of strain waterfall plots can be caused by many other factors; for example, non-orthogonal fracture development, or heterogeneous stress field, or activated existing faults. The findings from this study provide additional information for fracture diagnosis and help to interpret fiber sensor measurements more accurately.

## 6. Conclusions

An experimental investigation of low-frequency distributed acoustic sensing in a laboratory setting with two parallel fractures is presented. The experiment studied the offset strain response to two parallel propagating radial fractures, validated and revealed additional insights into our understanding of how fiber optic diagnostic tools can be utilized in the field. The findings of this work are listed below.
Strain measured by DAS for the dominating fracture of the two existing fractures exhibits the characteristics of a convergence pattern as reported for single fracture tests.The magnitude of extension between two competing fractures is stronger than the extension of a single propagating fracture.The zero-strain location method was applied to the experimental data and it accurately predicted the fracture radius and distance of the fracture front to the fiber using the strain data from the embedded fiber intersected by the fracture.Between the two propagating fractures, the bottom propagates faster and becomes dominant. This could be caused by the hydrostatic pressure effect on the net pressure at each fracture.

## Figures and Tables

**Figure 1 sensors-24-03880-f001:**
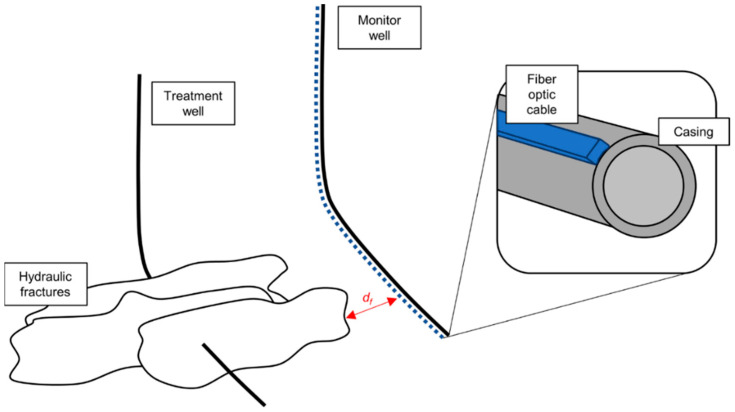
Schematic representation of a fiber optic sensor in a monitoring well [5]. A treatment well has fractures that propagate due to the high-pressure injection. A far-field monitoring well has a fiber optic sensor installed along the wellbore which senses the strain rate variation in the surrounding formation due to the fracture propagation. As fractures approach the monitoring well, the LF-DAS registers the strain rate change.

**Figure 2 sensors-24-03880-f002:**
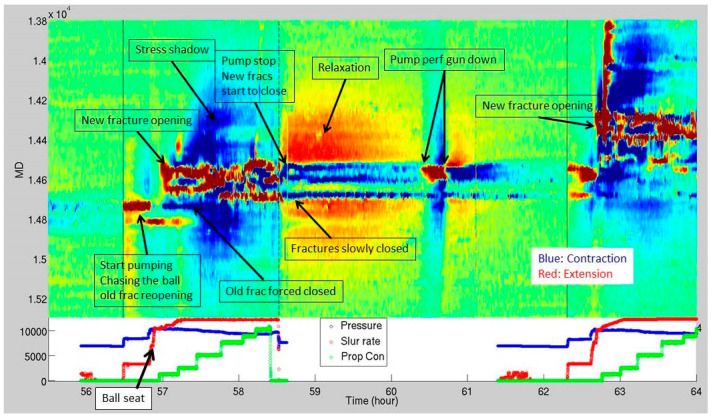
LF-DAS waterfall plot during hydraulic fracturing in offset well where the color represents strain (blue for compression and red for extension) (adopted from Raterman [9]).

**Figure 3 sensors-24-03880-f003:**
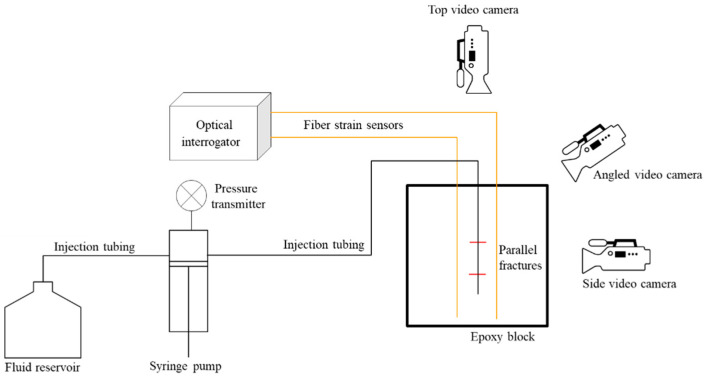
Schematic representation of the lab-scale hydraulic fracture experiment with two parallel fractures. Injection tubing serves as treatment well, and fiber sensors serves as monitoring wells. Pressure, injection rate, and fracture geometry are monitored during injection, in addition to strain monitoring by fiber sensor.

**Figure 4 sensors-24-03880-f004:**
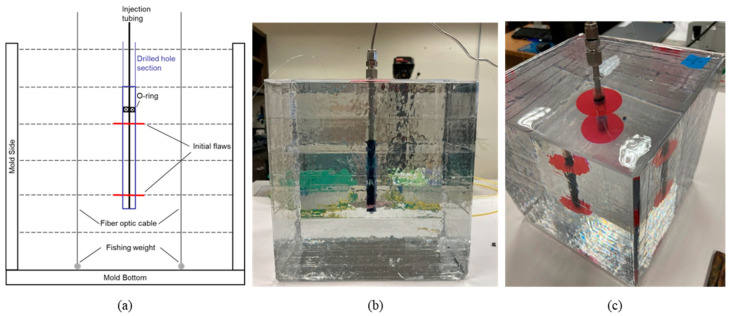
(**a**) Schematic representation of epoxy specimen of two parallel fractures from side view with two fiber optic cables; (**b**) picture from side view of fracture specimen before injection test; (**c**) picture from top angle view of fracture specimen before injection test.

**Figure 5 sensors-24-03880-f005:**
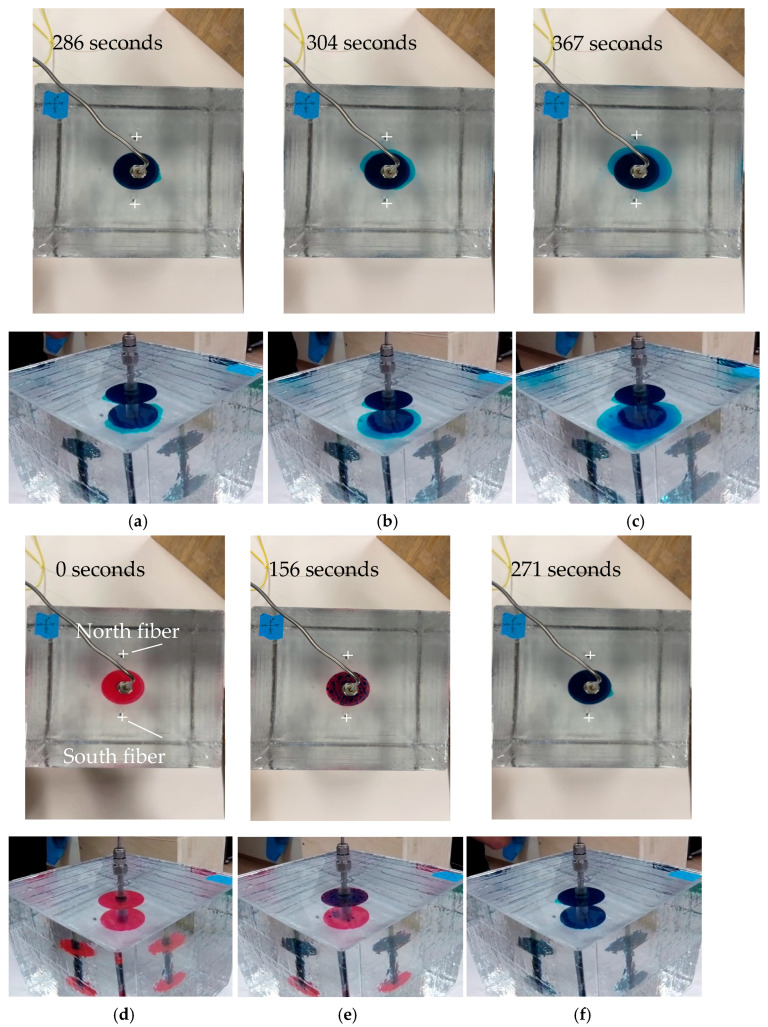
Time-ordered fracture geometry in fracture specimen: (**a**) initial, (**b**) top-fracture filled, (**c**) bottom-fracture filled, (**d**) both fractures propagate, (**e**) bottom fracture over-grow, and (**f**) bottom fracture dominates.

**Figure 6 sensors-24-03880-f006:**
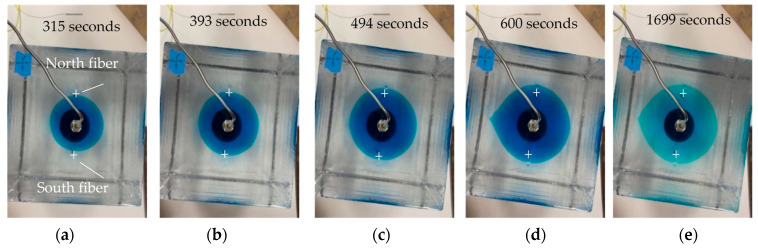
Top view of the specimen during experiment showing the fracture intercepting fiber sensors: (**a**) fracture hit north fiber, (**b**) fracture intercepted with both fibers, (**c**) fracture passed both fibers, (**d**) fracture breakthrough the block, and (**e**) experiment complete.

**Figure 7 sensors-24-03880-f007:**
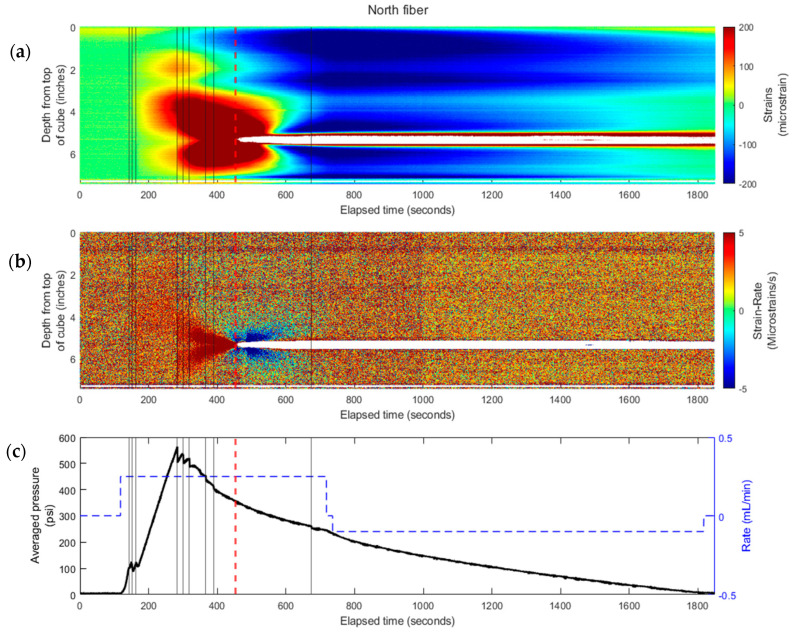
Experimental results for south fiber: (**a**) strain waterfall plot from fiber measurements and (**b**) calculated strain-rate waterfall plots—for a propagating fracture intersecting the embedded north fiber (see Figure 5) at 454 s. (**c**) The injection rate (blue dashed line) and averaged pressure (solid black line) profiles are plotted at the corresponding time during the experiment. The black vertical lines indicate fracture growth and the red dotted line indicates when fracture intercepts the fiber.

**Figure 8 sensors-24-03880-f008:**
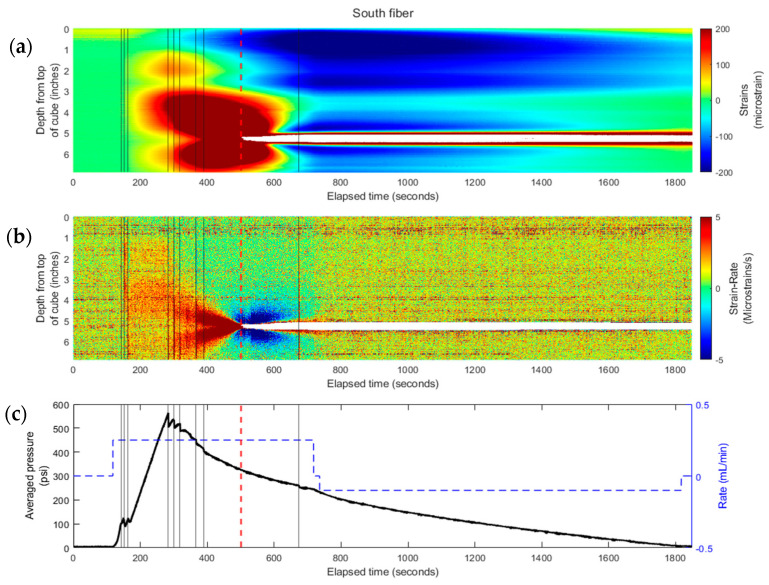
(**a**) DAS recorded strain waterfall plot and (**b**) calculated strain-rate waterfall plot—for a propagating fracture intersecting the embedded south fiber (see Figure 5) at 501 s. (**c**) The injection rate (blue dashed line) and averaged pressure (solid black line) profiles for the south fiber. The black vertical lines indicate fracture growth and the red dotted line indicates when fracture intercepts the fiber.

**Figure 9 sensors-24-03880-f009:**
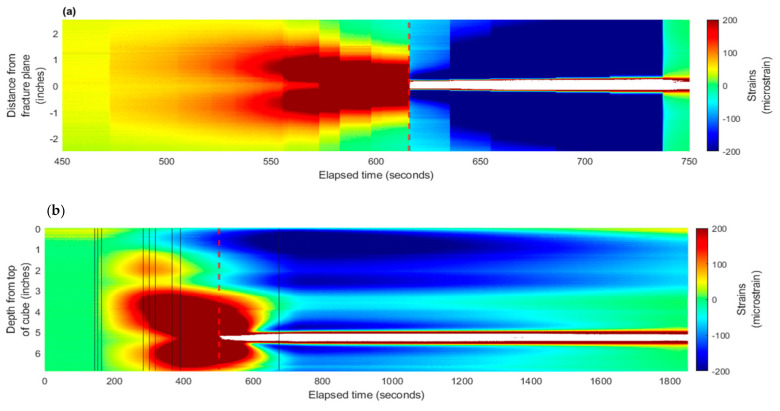
Comparison of strain rate plot for one-fracture system and two-fracture system. (**a**) single fracture, (**b**) two fractures.

**Figure 10 sensors-24-03880-f010:**
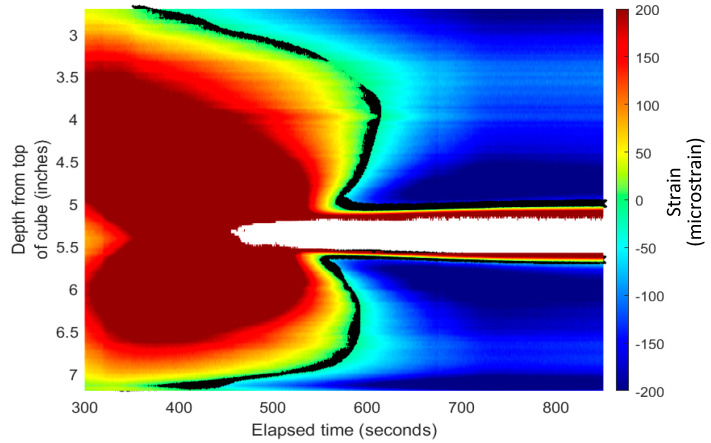
Strain waterfall with marked zero-strain locations (black line) where tension (fracture propagation) become compression (non-fractured rock).

**Figure 11 sensors-24-03880-f011:**
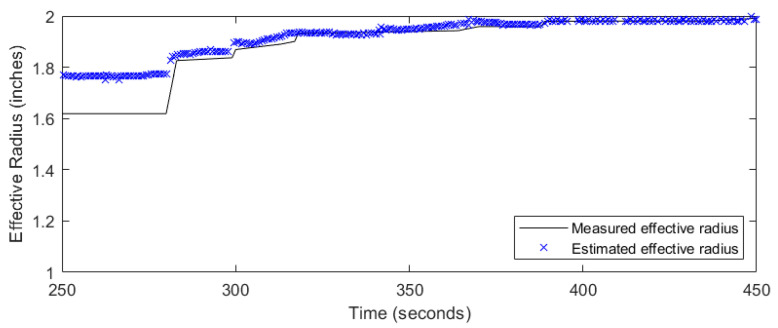
Measured and estimated fracture radii using the zero-strain location method.

**Figure 12 sensors-24-03880-f012:**
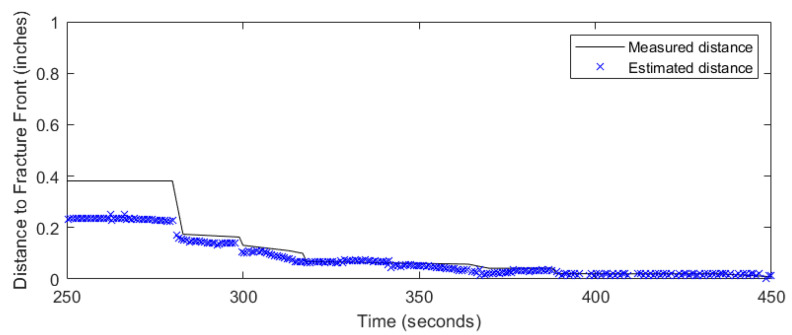
Measured and estimated distance to fracture front using the zero-strain location method.

**Figure 13 sensors-24-03880-f013:**
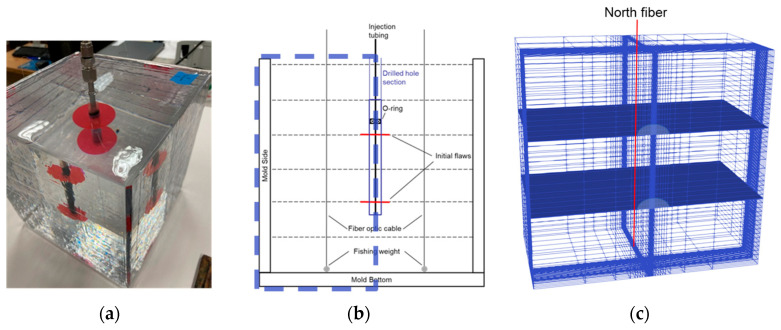
(**a**) Picture from top angle view of fracture specimen before injection test; (**b**) schematic representation of epoxy specimen of two parallel fractures from side view with two fiber optic cables; (**c**) finite-element model domain with north fiber location in red line.

**Figure 14 sensors-24-03880-f014:**
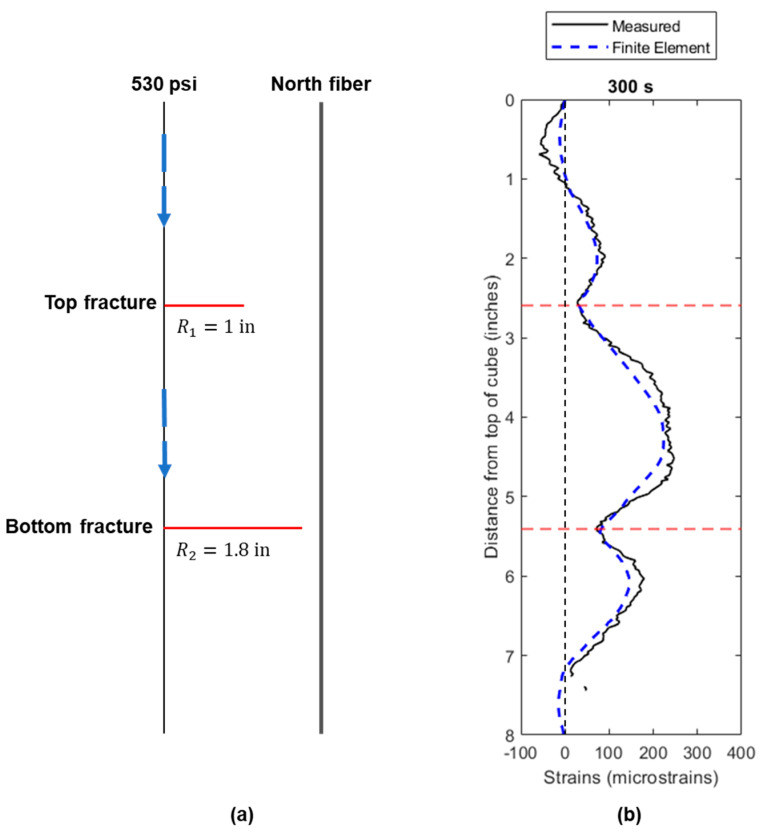
(**a**) Schematic representation of two parallel fractures from side view with north fiber optic cable; (**b**) comparison of measured and finite element modeled strains at 300 s.

**Table 1 sensors-24-03880-t001:** Zero-strain curve fit coefficients [1].

	*Z*_0*D*_ > 1	0.75 ≤ *Z*_0*D*_ ≤ 1	*Z*_0*D*_ < 0.75
*a*	−0.141ν − 0.473	−1.23ν − 0.702	−4.06ν − 3.01
*b*	0.0707ν + 0.00106	1.72ν + 0.142	7.82ν + 3.73
*c*	0.331ν + 1.05	−0.246ν + 1.04	−3.39ν − 0.348

**Table 2 sensors-24-03880-t002:** Performance specifications of the operational parameters for the fracture injection experiment [15].

Parameter	Specification *
Strain measurement range	±15,000 µε
Resolution	0.1 µε
Instrument accuracy	±1 µε
System accuracy	±25 µε
Measurement uncertainty at zero strain	±5 µε

* µε represents the microstrains measured by the interrogator unit.

## Data Availability

Data are contained within the article.

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
