# Peer review of "Experimental Investigation of Low-Frequency Distributed Acoustic Sensor Responses to Two Parallel Propagating Fractures†"

_sensors, 2024, doi:10.3390/s24123880_

Round 1

Reviewer 1 Report

Comments and Suggestions for Authors

This paper presents a detailed experimental investigation of low-frequency distributed acoustic sensing (LF-DAS) techniques for monitoring fracture propagation in hydraulic fracturing. The research holds significant practical value for the oil and gas industry and related geological engineering sectors. Some specific feedback on your paper is provided below:

1.    In the "Strain Measurements" section, while distributed HD fiber optic strain sensors are mentioned, the operational principles and technical specifications of these sensors are not sufficiently explained. To enhance the article's integrity and transparency, it is suggested that the author provide more detailed descriptions of the technical aspects of the OFDR sensor.

2.    The paper mentions the growth and competition of cracks during the experiment; however, the specific mechanisms and influencing factors of crack growth are not explored in depth. It is recommended that the authors delve further into the dynamics of fracture growth and illuminate how these factors impact the monitoring outcomes of LF-DAS in the discussion section.

3.    Figure 1 of the paper presents a schematic diagram of the optical fiber in the monitoring well, though the specific arrangement and function of the optical fiber during the experiment are not clearly elucidated. It is advisable that the authors incorporate detailed information about the fiber layout in the caption of Figure 1.

4.    In the experimental section, the use of a transparent cube of epoxy resin to simulate formation rocks is mentioned; however, the specific mechanical properties of epoxy resin, such as Poisson's ratio and Young's modulus, are not provided. These parameters are vital for comprehending the experimental results. It is recommended that the authors include this information in the Materials and Methods section.

5.    In the conclusion, the main findings of the experiment are summarized, but there is insufficient discussion on how these findings are applied to actual hydraulic fracturing operations and completion design optimization. The author is encouraged to further explore the practical engineering implications of these experimental results in the conclusion.

6.    While several previous studies are cited in the article, the correlation and comparative analysis of these works with the present study are insufficiently described. It is advisable that the author include a review of the relevant literature in the introduction or discussion section, particularly where the results of this study are compared and contrasted.

7.    Page 7 contains errors in the layout and serial numbering of the experimental pictures; on page 8, there is an error in the description of Figure 6; on page 14, the serial number of the figure is incorrect, and correction is needed.

Author Response

Please see the attached for responses to the comments.

Thanks for your time and suggestions.

Reviewer 2 Report

Comments and Suggestions for Authors

The authors report a very interesting study of the hydraulic fracture process monitored by DAS system. Dynamic evolution of the fracture process is successfully recorded by the sensor. The measurement results are in good agreement with the modelling. The work can be published in Sensors, I only have some minor issues.

1) Fig. 7(a) and 8(a) are a bit difficult to follow. It is better to directly mark the position of the fiber in the figure. 

2) It is better to plot Fig. 9(a) and (b) in the same scale in time, such that to compare the differences between them.

3) I would suggest to highlight in the text why low-frequency DAS is required in this application, and what is frequency range the authors are targeting at. 

Author Response

(The authors gave the same response as above.)

Reviewer 3 Report

Comments and Suggestions for Authors

I have carefully read the manuscript entitled "Experimental Investigation Low-frequency Distributed Acoustic Sensor Responses of Two Parallel Propagating Fractures" by Reid et al., and I believe that Moderate/Major revisions are required mainly in the organisation of the paper structure. Specifically, I find that the submitted study is lacking novelty and it is more of a continuation of previously published work. Therefore, in order to be accepted for publication the gap of knowledge that the current study is attempting to fill in, should be made crystal clear in the Introduction. Moreover, I find it bad practice to include a "Results and Discussion" section, and I would rather prefer to see two distinct sections in the revised manuscript. My main and minor comments are included in the attached pdf file.

Author Response

please see the attached for responses to the comments. The responses are embedded in the document.

Round 2

Reviewer 1 Report

Comments and Suggestions for Authors

This paper focuses on the application of Low-Frequency Distributed Acoustic Sensing (LF-DAS) technology for monitoring fracture propagation during hydraulic fracturing.  This work has significant practical implications and valuable applications in the oil and gas industry as well as related geotechnical engineering fields.  Overall, the paper is well-written with a clear presentation of the key ideas.  It is acceptable after revisions.

In the current version, there are a few issues that need clarification.  Although the paper is structured well and contains rich content, it includes several formatting errors that need to be addressed.  For instance, there is an incorrect citation in the explanation of equation 10.  Additionally, on page 15, the last sentence incorrectly cites a number.  Please thoroughly check and correct these issues as well as any other potential formatting discrepancies.

Author Response

Eq. 10, problem is fixed

Mistake on page 15 was corrected.

Thank you for your review comments.

Reviewer 3 Report

Comments and Suggestions for Authors

The authors attempted to address some of my comments in my first review, but I feel that a couple of issues still need to be improved.

More importantly, I feel that this paper does not make any connection in the Discussion of the findings from a laboratory test to real case studies. I would suggest to the authors to add references and new text in the Discussion supporting their lab findings or discussing any divergence in their results from findings from real strain monitoring studies in wells. The paper in its form is not very conclusive.

I personally do not find good practice to keep the Results and Discussion sections together, but I understand that this may be a way to avoid repetitions in the manuscript, so I will not insist to this comment in my initial review, but I would certainly encourage the authors to avoid this.

There are two Figures numbered as Fig. 9. Please correct the numbering

Fig.9 Please add a label next to the colour scale. Is this strain?

Author Response

  1. the comments on field application. The method presented cannot be used as standalone interpretation tool. It adds on to the previous work on estimate the distance between fracture front and near-by existing wells to prevent frac-hit which causes problems on production performance and casing deformation. This paper serves one purpose, identify the features that alter the standard characteristics of DAS measurements when more than one fractures exist. We have added one paragraph before the conclusion to explain how to use this method.
  2. We have fixed problem of 2 Fig. 9. Thanks for catch that error.
  3. We have added the legend to Fig. 9.